# The impact of childcare availability on maternal employment: Evidence from Czech municipalities

**Klára Kalíšková**[1,2]*, **Daniel Münich**[1]

**1** CERGE-EI, A Joint Workplace of Charles University in Prague and the Economics Institute of the Czech Academy of Sciences, Prague, Czech Republic, **2** Faculty of Informatics and Statistics, Prague University of Economics and Business, Prague, Czech Republic

* klara.kaliskova@cerge-ei.cz

## Abstract

This study presents empirical estimates of the effects of local public kindergarten availability on the employment rate of mothers of preschool-aged children in the Czech Republic. It brings together data from the Population Census and School Register to form a unique database describing the demand for childcare (numbers of children aged 3–5) and supply (capacity) across more than 6,000 Czech municipalities between 2001 and 2011. We take advantage of variations in kindergarten availability over time to estimate the impact on maternal employment with a first differences model. Our estimates show that a 10 percentage point increase in the availability of kindergarten places led to a 0.2–0.4 percentage point increase in the employment rate of mothers of preschool-aged children. A drop in the availability of kindergarten places between 2001 and 2011 resulted in 9,000 fewer mothers being employed and a net loss to public finances of 1.2–1.7 billion CZK per year.

## 1. Introduction

Low employment rates of mothers of pre-school children remain a significant barrier to narrowing gender gaps in the labor market [1]. The availability of public childcare has long been recognized as an important and obvious determinant of maternal employment, but recent studies on the impact of childcare availability have produced mixed results (for a review of this literature, see [2]). In Western Europe and the US, the estimated impact of the availability of public childcare on maternal employment ranges from none or very small in the Netherlands, England, the US, Norway, and France [3–9], to medium and large in Germany, Italy, and Spain [10–12].

Overall, the effectiveness of available public childcare in increasing maternal employment appears to crucially depend on the institutional setting in each country, including prevailing social norms, family policies, and the availability of flexible forms of work [2]. These are usually strongly correlated to pre-existing maternal employment rates and thus a pattern has emerged in this literature. Increases in public childcare capacity appear to have a strong impact on maternal employment rates in countries with previously low rates, while the impact is negligible or non-existent in countries where high maternal employment rates are the norm.

**PLOS ONE**

**Data Availability Statement:** We have uploaded the minimal anonymized data set and related codes necessary to replicate our study findings to Zenodo: https://doi.org/10.5281/zenodo.7989063.

**Funding:** This paper is based on a policy report "Analýza dopadů (ne)dostupnosti míst v mateřských školách na participaci žen na trhu práce" supported by the Czech Ministry of Labor and Social Affairs from the European Social Fund project CZ.03.1.51/0.0/0.0/15_009/0003702. The academic version of the article was developed with the support of the NPO "Systemic Risk Institute" number LX22NPO5101, funded by European Union - Next Generation EU (Ministry of Education, Youth and Sports, NPO: EXCELES) and by the Czech Academy of Sciences as part of its AV21 Strategy programme. The funders had no role in study design, data collection and analysis, decision to publish, or preparation of the manuscript.

**Competing interests:** The authors have declared that no competing interests exist.

Until recently, evidence on the impact of the availability of public child care on maternal employment was limited in Central and Eastern European (CEE) countries, which are known to have relatively high female labor force participation and primarily publicly provided childcare. In two recent contributions, [13, 14] estimate a much larger effect of the availability of public childcare in Hungary and Poland than was found in studies of Western European countries and the US. The only comparable estimates come from a 1996 introduction of children's legal claim to a place in kindergarten in Germany, where [10] found similarly large effects on maternal employment as those found in Hungary and Poland. [15] confirm this conclusion. They provide a direct comparison of the impact of childcare availability when the child is around the age of 3 on maternal employment in selected Central and Eastern European countries, and in Western and Southern European countries. They conclude that the impact of eligibility for subsidized childcare on maternal employment is the largest in CEE countries, while it is smaller and not long-lasting in Western EU countries, and insignificant in Southern Europe.

Evidence from Central and Eastern Europe can considerably extend knowledge in this field, and offer unique insights from an interesting institutional setting. We contribute to the limited evidence on the impact of childcare availability on maternal employment in CEE countries by providing estimates from the Czech Republic. We utilize notable dynamic demographic changes in terms of children born, non-uniformly distributed across municipalities, which are the major providers of kindergarten services. We also utilize the fact that the Czech Republic has an extraordinarily large number of self-governed municipalities (the average population size of Czech municipalities is among the lowest within OECD countries). We estimate the impact of the availability of public kindergarten on employment of mothers of 3- to 5-year-olds using this unique municipal-level variation.

We estimate that a 10 percent increase in the availability of public kindergarten leads to a 0.2–0.4 percentage point increase in the employment rate of mothers of 3- to 5-year-olds. Our results are lower than previous evidence from CEE countries, but are quite comparable to estimates from some Southern and Western European countries. For example, [16] estimated that a 10% increase in child care availability in Italy increases the relative odds of maternal employment by 0.296. Recent evidence from Hungary and Poland suggests that increasing childcare availability by 10 p. p. increases maternal employment by 1.17 p. p. [13] and 4.2 p.p. [14]. To the best of our knowledge, there is only one study that estimates the causal impact of childcare availability on maternal employment in the Czech Republic–[15] provides estimates for three CEE countries–the Czech Republic, Hungary, and Slovakia. Their estimates are very similar for all three countries and range from 0.07 to 0.09, i.e., still somewhat larger than our results, but lower than the studies from Hungary and Poland mentioned earlier. However, all previous studies on CEE countries use a different methodology than ours–they take advantage of childcare eligibility cutoffs [13, 15] and a change in the primary school starting age [14]. We use municipal-level variation in availability of kindergarten places. Moreover, unlike [15], who focus on children born around the childcare eligibility cut-off, our estimates include all mothers with a youngest child aged 3–5.

In addition to our estimates of the impact of childcare availability on maternal employment, we also calculate the aggregate effects and fiscal impacts of changes in kindergarten capacity by taking the size of the affected populations into account. Over the period we study, the national deficit in local kindergarten capacity increased by two thirds, from 63,027 missing places in 2001 to 105,771 in 2011. This deficit of almost 43 thousand kindergarten places corresponded to approximately 36% of the cohort of 3-year-old children in 2011. This resulted in 9,000 fewer mothers being employed, which is 5% of all mothers whose youngest child was between 3 and 5 years old in 2011. This reduction in kindergarten availability reflects a net loss to the public

finances of 1.2–1.7 billion CZK per year due to the decrease in employment of mothers of young children.

While the availability of kindergarten places improved in the Czech Republic around the year 2016, the deficit in kindergarten capacity increased again in recent years. The number of rejected applications for kindergartens started to rise during COVID-19 pandemics and the situation of parents with pre-school children was further worsened by one of the longest closures of kindergartens that took place during the COVID-19 pandemics in the Czech Republic [17]. While our estimates of the impact of the unavailability of kindergarten places on maternal employment do not use data from these recent years, we believe that our estimates of the impact of childcare availability on maternal employment are still informative for the current situation. The responsiveness of maternal employment towards childcare availability seems to be mostly driven by differences in cultural and institutional contexts and the pre-existing maternal employment rates [2] rather than showing substantial changes over time within one country.

The rest of the paper is organized as follows: the next section describes childcare institutions in the Czech Republic, section 3 introduces our data and methodological approach, section 4 provides our regression results, aggregate impacts, and discussion of our findings, and section 5 concludes.

## 2. Institutional background

Childcare in the Czech Republic has two main pillars: public nurseries for children aged 1–2 and public kindergartens for children aged 3–5. The numbers of nurseries, which operated as medical facilities under the jurisdiction of the Ministry of Health, were substantially reduced in the 1990s and virtually ceased to exist by the early 2000s. 15% of children aged 1–2 were enrolled in nurseries in 1990, falling to just under 1% by the end of the 1990s (Institute of Health Information and Statistics of the Czech Republic). In 2011, nurseries ceased to exist as medical facilities (Act No. 372/2011).

Private childcare is very rare in the Czech Republic for children both under and over 3. While no administrative data on private childcare facilities exist, [18] surveyed a representative sample of Czech municipalities in 2002, and found that only a few offered any form of private childcare (of 497 villages in the survey, there were 3 private kindergartens, 8 childcare agencies, and 4 self-help organizations focused on childcare). Childminders are employed by only about 2% of parents with children aged 1–10 and they are generally employed only occasionally, not for daily childcare [19]. It is more common for grandparents or other relatives to help with children. 20% of parents with children aged 1–10 mentioned that grandparents help them with childcare on a regular basis during weekdays, but the help mostly constituted of picking-up children from kindergarten or school, not full-time care [19]. Therefore, the share of children under 3 in formal care is persistently low in the Czech Republic–in 2005 (the first year for which data is available), only 2% of children aged 0–2 were enrolled in formal childcare. The Czech Republic has the lowest share of children under 3 in formal childcare in the whole European Union; see Fig 1. This has not changed substantially over time (in 2015, the share was 2.9%; see Fig 2).

The primary caretakers of children under 3 in the Czech Republic are their mothers. This is supported by generous paid parental leave policy, which offers job-protected leave until the child's 3rd birthday and a flat rate parental allowance that can be collected until the child is 4. The parental allowance amounted to about one fourth of the average wage in the economy in the 2000s (the amount was valorised annually). Since 2008, the parental allowance system has been made more flexible. Some parents (with sufficiently high pre-birth earnings) became

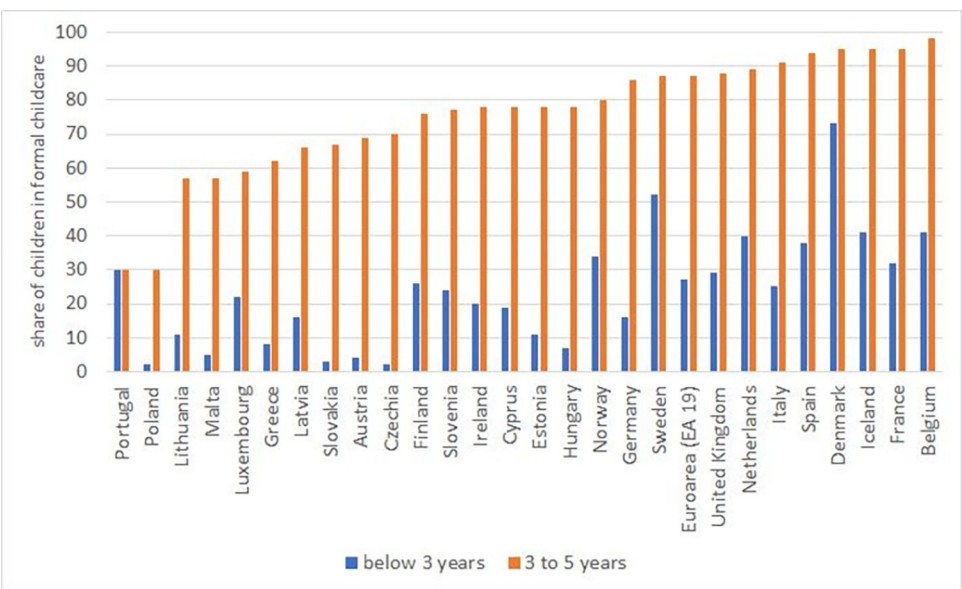

**Fig 1. Share of children in formal childcare by age, 2005.** Note: The blue columns report the share of children in formal childcare for those under 3, while the orange columns report the same statistic for children aged 3–5 years (or until compulsory school starting age). Source: Eurostat, Children in formal childcare or education by age group and duration: https://ec.europa.eu/eurostat/databrowser/view/ILC_CAINDFORMAL__custom_2178651/default/table?lang=en.

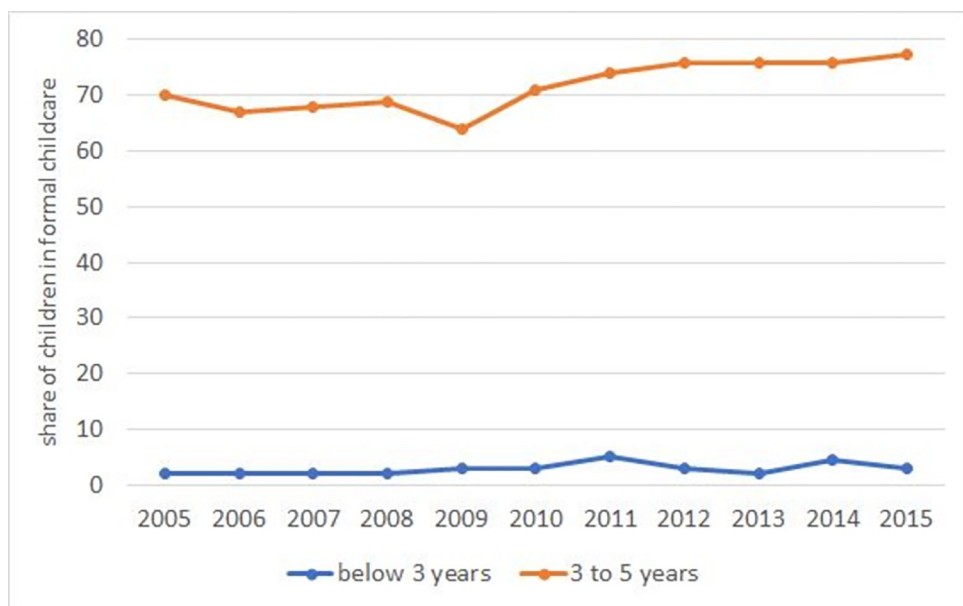

**Fig 2. Share of children in formal childcare by age in the Czech Republic.** Note: The blue line reports the share of children under 3 in formal childcare, while the orange line reports the same statistic for children 3–5 (or up to the compulsory school starting age). Source: Eurostat, Children in formal childcare or education by age group and duration: https://ec.europa.eu/eurostat/databrowser/view/ILC_CAINDFORMAL__custom_2178651/default/table?lang=en.

entitled to choose a leave shorter than 4 years (the choice is between 2, 3, and 4 years) with a corresponding (higher) monthly parental allowance. In principle, the parental allowance can be received by a father, but this is very rare (1.8% of recipients were fathers in 2015).

When children reach the age of 3 and job-protected parental leave ends, parents often apply for a place in a public kindergarten. Kindergartens operate within the jurisdiction of the Ministry of Education, Youth, and Sports, and constitute the main pillar of the pre-school educational system for children aged 3–5. Compulsory basic school attendance starts on September 1 after a child's 6th birthday. Children who are not sufficiently physically or mentally mature can postpone their school starting age (conditions for such postponement are defined by law), so there can be children older than 6 who attend kindergartens.

Kindergartens are largely subsidized by the government and are affordable for the vast majority of parents—the average fee constituted only 2.7% of the net income of an average family in 2006 [18]. Given that private childcare is very rare and much more expensive, most children 3–5 attend public kindergartens. In 2005, 70% of children between 3 and 5 were enrolled in some institutional childcare, which is slightly below the EU average (see Fig 1). This share increased just slightly to 74% in 2011 and then continued to slowly rise afterwards (Fig 2).

Admissions criteria are set by individual kindergartens, but priority is always given to children who are registered as permanently resident in the given municipality (and who meet the age criteria). Sometimes, kindergartens further restrict their admission criteria to only children whose parents both work, or they give priority to children who have a sibling in the same kindergarten [18]. Children who live in municipalities that have no kindergarten or who are not admitted to their local kindergarten due a lack of capacity can apply to kindergartens in nearby municipalities, though strict priority is given to children who live in that municipality.

The capacity of public kindergartens does not meet the demand. More than 13 thousand applications for kindergarten enrollments were declined in 2007, the first year for which this data is available (source: Czech Ministry of Education, Youth, and Sports). This amounts to 4.8% of children aged 3–5. The share of declined applications on the population of all children aged 3–5 increased in the following years, reaching almost 15% in 2011. This suggests that the size of the excess demand likely increased over time. While this is the best measure of excess demand that we have, it is certainly not perfect. Parents can apply for an enrollment in more than one kindergarten, so each declined application does not necessarily correspond to one child. On the other hand, if parents know that there are no openings in kindergartens in their neighborhood, they may not even apply. The measure could thus be biased both upwards and downwards.

Importantly, there are large regional and municipal differences in availability of kindergarten places [18]. These stem from two sources. The first source of variation comes from the notable dynamic demographic changes in the 2000s. In the 1990s, there was a steep decline in birth rates in the Czech Republic (as well as in other post-communist countries, see e.g., [20]), but the next decade was marked by a pronounced increase in life births (see Fig 3). This was a consequence of the generation of baby boomers from the 1970s reaching childbearing age. However, the rise in birth rates was non-uniformly distributed across districts. See the cumulative changes in fertility rates in Czech districts in 2007–2009 and 2010–2012 depicted here (source: Czech Statistical Office): d796bc59-b416-4aea-955c-895ed93c1c3f (czso.cz); 56fb920c-098d-437d-8d3b-c396a86f4010 (czso.cz).

The second source of variation arises from the fact that kindergartens are established by municipalities, and there is an extraordinarily large number of municipalities in the Czech Republic. The average number of inhabitants in a Czech municipality was fewer than 1,700 in 2016, the lowest municipal size in the EU [21]. While municipalities operate Czech public kindergartens, they are not obliged to establish them or to increase their capacity when excess

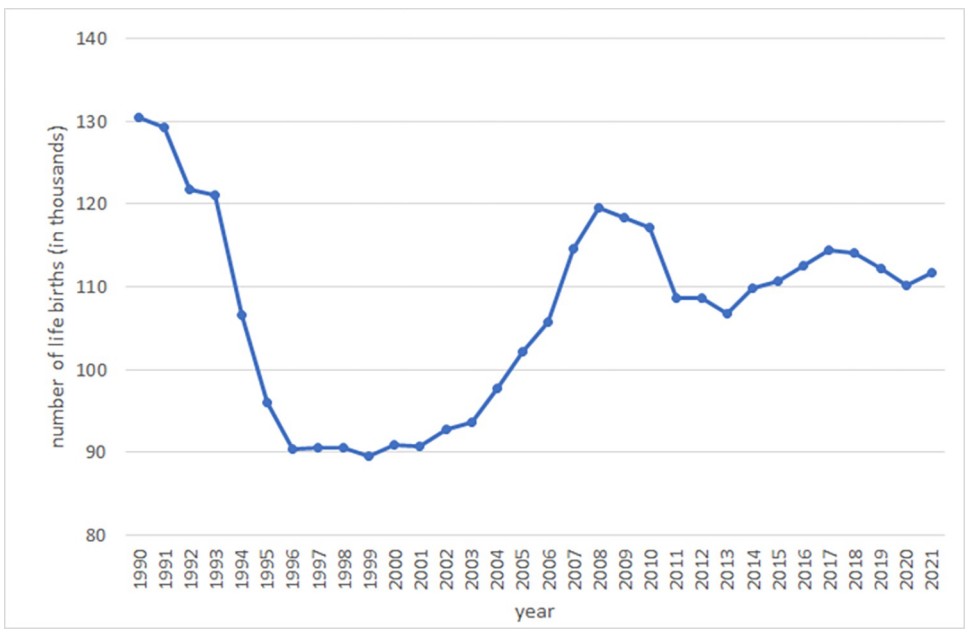

**Fig 3. Evolution of life births in the Czech Republic.** Note: The figure depicts number of life births in the Czech Republic 1990–2021. Source: Czech Statistical Office, https://www.czso.cz/csu/czso/obyvatelstvo_hu.

demand is present. If they decide to do so, municipalities must go through a rather complicated administrative process requiring, among other things, obtaining a subsidy from the central government on a very competitive basis. Municipalities must also cover all operational costs of a kindergarten, except pedagogical staff salaries. Further, in the 2000s, satellite towns were often built around large cities, and many lack sufficient infrastructure, including available childcare. Families with small children often moved into these satellite towns, creating further discrepancies between the demand for and the supply of kindergarten places [18].

Overall, changes in regional availability of kindergarten places are driven by both demographic changes in each municipality and the decisions of municipal councils regarding kindergarten capacity. We assume that these changes are thus exogenous to the decisions of mothers with pre-school children concerning their return to the labor market, and we use this variation to estimate the causal impact of kindergarten availability on the maternal labor supply (see Section 3.3. for a discussion of these identification assumptions).

## 3. Methodology

### 3.1. Data and definition of variables

Our measure of maternal employment uses partially aggregated data from the Census of Population, Buildings, and Dwellings, carried out by the Czech Statistical Office in 2001 and 2011. Data from the 2021 Census were not available at the time this manuscript was written. Our sample includes all women whose youngest child is aged 3 to 5. We do not further categorise the women according to the age of their youngest child, because this would yield several categories with zero counts in numerous municipalities. We distinguish between 12 demographic categories of mothers: by number of children (3 categories: 1, 2, or 3 and more children) and by level of education (4 categories, described below). Unfortunately, the data do not have information on marital or cohabitation status of women, so we cannot do the analysis separately for single and cohabiting/married women.

For each Czech municipality, each year of observation (2001 and 2011), and each group of women defined by their characteristics (level of education and total number of children), we calculate the number of mothers who are in employment as a proportion of the total number of mothers with the same characteristics. This gives our main outcome variable–the maternal employment rate for women with a youngest child aged 3–5 measured at year-municipality-group level. The employment rate among mothers of 3–5 years olds increased over the period, from 57.8% in 2001 to 68.8% in 2011—a change of 11 percentage points, which corresponds to a change in the employment rate of 19% (Table 1). Changes in maternal employment rates at the municipality level from 2001 to 2011 range between the extreme values of -1 and 1. The value -1 corresponds to a situation in which all mothers of kindergarten-aged children in the given municipality were employed in 2001, but in 2011, none were. Naturally, these extreme changes occurred only very rarely (Fig 4). Changes in the employment rate close to zero were more common, and most often are in positive figures, consistent with an increase in the overall employment rate across the period.

At the same time, the educational structure changed in favor of more highly educated mothers (10.2% had only primary education in 2001, compared with 7% in 2011, and mothers with tertiary education increased from 9.8% to 17.3%; see Table 1). This is in line with a documented strong growth trend in the share of tertiary educated women in the Czech Republic beginning in the first decade of the 2000s. Meanwhile, family structure in terms of numbers of children shifted in favor of smaller families (37% of women had only one child in 2001, against 40.8% in 2011, while the number of families with three and more children fell from 15.8% to 11.2%; see Table 1).

Data on the capacity of public kindergartens at the municipality level are taken from the School Register maintained by the Ministry of Education, Youth, and Sports of the Czech Republic. In total, we have observations in both years for 6,098 municipalities. We exclude 19 municipalities from the sample because their boundaries (and often names) were changed during the observed period. Altogether these made up 0.03% of the total sample (they were municipalities with a few inhabitants).

The availability of kindergartens is calculated as a ratio of the number of kindergarten places to the number of children in the target ages. Such a measure is commonly used in the literature to proxy childcare availability [11, 14, 16, 22]. However, most studies use a higher level of aggregation to calculate this measure: while we take advantage of municipal-level data, most studies have data only at a regional or provincial level. While the lower level of aggregation in our study provides an undisputable advantage for our estimation, it also poses a challenge, because actual childcare availability might also depend on availability outside of one's municipality of residence. While priority in admissions is always given to children who reside in the given municipality, parents can also apply to kindergartens in nearby municipalities, as explained in Section 2. Therefore, we use two indicators to capture the availability of kindergartens: local and neighboring kindergarten availability.

### Local availability of kindergarten places (LA)

*Local availability of kindergarten places (LA)* captures the availability of places in the municipality where a child lives. It is expressed as the total number of kindergarten places (kindergarten capacity) in the given municipality as a share of the total number of 3- to 5-year-old children resident in the municipality at the time. The numbers of children aged 3–5 resident in each municipality also come from the Census of Population, Buildings, and Dwellings. For municipalities with no kindergarten, the value of this index is zero. For index values greater than one (where there are more kindergarten places in the municipality than preschool-aged

**Table 1. Main summary statistics.**

| year | | EMPL | LA | NA1 | NA3 | NA5 | pri | sec1 | sec2 | ter | children1 | children2 | children2 | Number of observations |
|------|------|------|------|------|------|------|------|------|------|------|------|------|------|------|
| | | Employment rate among mothers | Local kindergarten availability | Neighboring kindergarten availability (1 nearest) | Neighboring kindergarten availability (3 nearest) | Neighboring kindergarten availability (5 nearest) | Share of mothers with primary education | Share of mothers without school leaving certificate | Share of mothers with secondary school leaving certificate | Share of mothers with tertiary education | Share of mothers with 1 child | Share of mothers with 2 children | Share of mothers with 3 + children | |
| 2001 | mean | 0.578 | 0.829 | 0.334 | 0.174 | 0.108 | 0.102 | 0.421 | 0.380 | 0.098 | 0.370 | 0.472 | 0.472 | 197 359 |
| | std. dev. | 0.494 | 0.271 | 0.42 | 0.25 | 0.16 | 0.302 | 0.494 | 0.485 | 0.297 | 0.483 | 0.499 | 0.365 | |
| 2011 | mean | 0.688 | 0.766 | 0.182 | 0.077 | 0.048 | 0.070 | 0.284 | 0.473 | 0.173 | 0.408 | 0.48 | 0.112 | 173 851 |
| | std. dev. | 0.463 | 0.26 | 0.338 | 0.149 | 0.085 | 0.255 | 0.451 | 0.499 | 0.378 | 0.491 | 0.5 | 0.315 | |
| Change between 2001 and 2011 | | | | | | | | | | | | | | |
| | | ΔEMPL | ΔLA | ΔNA1 | ΔNA3 | ΔNA5 | | | | | | | | |
| | mean | 0.066 | -0.062 | -0.147 | -0.093 | -0.057 | | | | | | | | 160 336 |
| | std. dev. | 0.253 | 0.166 | 0.395 | 0.218 | 0.142 | | | | | | | | |

Source: Census of Population, Buildings, and Dwellings; School Register (2001 and 2011), Author's calculations.

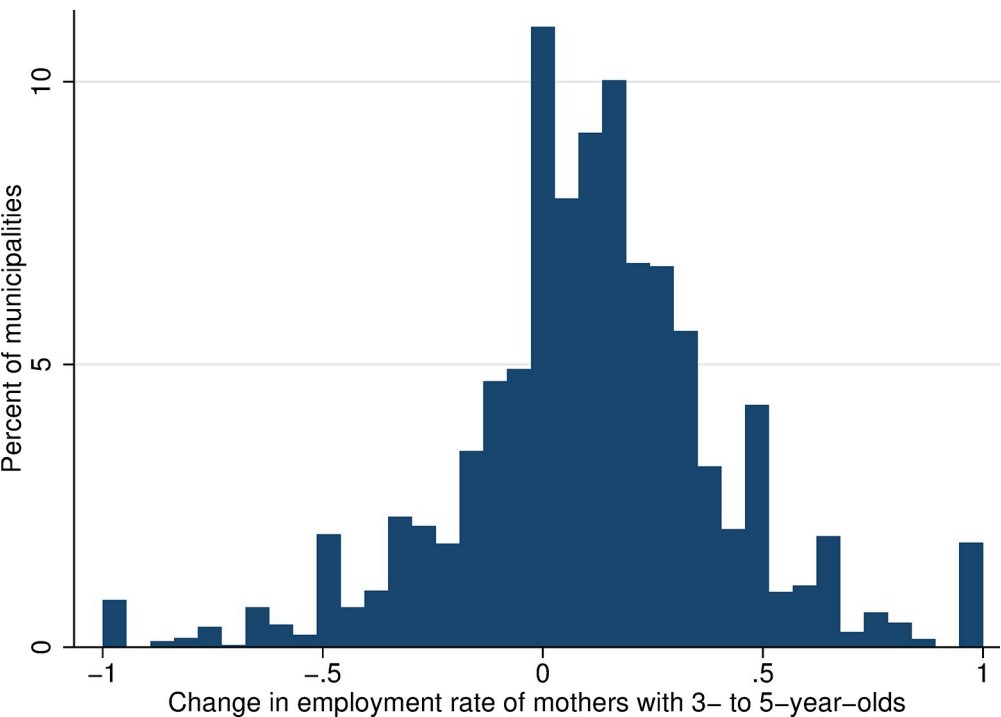

**Fig 4. Change in employment rates of mothers with 3- to 5-year-old children.** Source: Census of Population, Buildings, and Dwellings (2001 and 2011), Author's calculations.

children) we use the value 1 (i.e., maximum availability). Values between 0 and 1 then represent the probability of obtaining a kindergarten place in the given municipality.

The average availability of local kindergarten places fell between 2001 and 2011. Average availability was 0.829 in 2001 (meaning that, on average, there were places in public kindergartens for about 83% of local 3- to 5-year-olds), while in 2011 availability fell to 0.766 (Table 1). Standard deviation of this index is relatively high, at 0.27 in 2001 and 0.26 in 2011 (Table 1). The distribution of local kindergarten availability is highly concentrated at the extremes (Fig 5). Nearly half of all municipalities had no kindergartens in 2001 nor in 2011. At the same time, many municipalities had local availability of 1, meaning that the kindergarten capacity is sufficient—in some cases, more than sufficient—to provide for all local children aged 3–5. This was the case for about 35% of all municipalities in 2001, and slightly fewer (about 28%) in 2011 (Fig 5). In the remaining municipalities, local kindergarten availability ranges between zero and full capacity, however, most are closer to full capacity (the availability is usually between 0.7 and 1, which corresponds to places for between 70 and 100% of local 3- to 5-year-olds).

Focusing on within-municipality changes in the availability measure, in the majority of municipalities (69%), there was no change in local kindergarten availability between 2001 and 2011 (Fig 6). In the remaining municipalities, there were either small changes in availability, on the order of plus/minus few percentage points or, on the contrary, there were absolute changes, i.e., an increase in local availability from 0 to 1 (reflecting establishment of a new kindergarten, in value terms $\Delta LA = 1$) or a decrease from 1 to 0 (closure of a local kindergarten, in value terms $\Delta LA = -1$).

The construction of this local availability index reflects the fact that strict priority in allocation of local kindergarten places is given to children who are registered as permanently

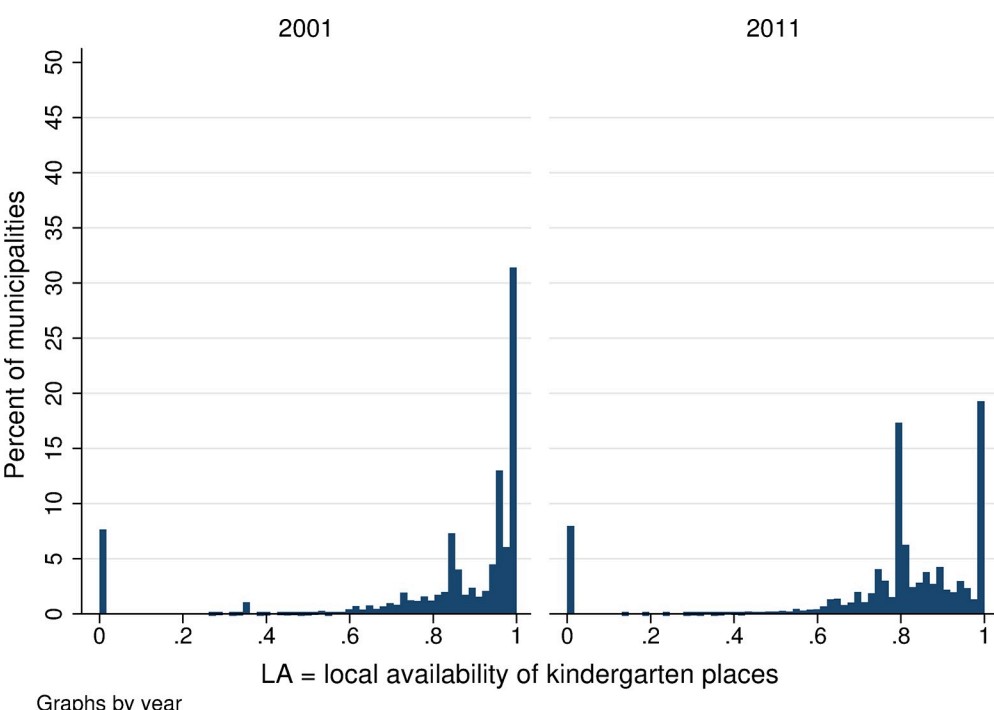

**Fig 5. Distribution of local availability of kindergarten places.** Source: Census of Population, Buildings, and Dwellings; School Register (2001 and 2011), Author's calculations.

resident in the given municipality, though children can apply in neighboring districts (see Section 2). We further assume that when parents do apply to a kindergarten in a neighboring municipality, they prefer kindergarten/s that is/are closest to their residence. We define the closest municipality with non-zero kindergarten capacity as the one at the shortest travel distance by car. Data on travel distances (by car) were obtained from Datlab s.r.o. We consider three scenarios–parents looking for a kindergarten in neighboring municipalities: 1. apply only to the one kindergarten that is closest to their residence; 2. apply to three kindergartens that are nearest to them; 3. apply to five nearest kindergartens.

**Neighboring availability of kindergarten places (NA).** We call the neighboring municipality/ies with a kindergarten *target* municipality/ies. The measure is calculated as the ratio of available places at the kindergarten in target municipality/ies where there is a kindergarten and demand for places. The available capacity (the numerator of this measure) is calculated as the difference between the total number of places in a kindergarten and the total number of resident children aged 3–5 in the target municipality/ies. The demand for these neighboring kindergarten places (the denominator) is defined as the sum of potential demand from all municipalities nearest to this municipality/ies. The demand is thus the sum of the differences between the number of children aged 3–5 and kindergarten capacity, summed over all municipalities that are nearest to specific target municipality/ies with kindergartens.

This is defined as:

$$NA \ (neighboring \ availability)$$

$$= \frac{kindergarten \ capacity \ in \ target \ municipality/ies - number \ of \ children \ aged \ 3-5 \ resident \ in \ target \ municipality/ies}{SUM[number \ of \ children \ aged \ 3-5 \ in \ nearest \ municipalities - kindergarten \ capacity \ in \ nearest \ municipalities]}$$

This is also the probability that a child will be successful in securing a kindergarten place in neighboring municipality/ies with a kindergarten. Because the denominator may be zero or negative, we limit the range of values of the index to the interval 0–1. For example, if there is zero demand for kindergarten places from nearby municipalities, we consider there to be maximum availability, i.e., 1. We calculate the neighboring availability of kindergarten places based on the three scenarios, which correspond to three variables. We assume that parents looking for a kindergarten in neighboring municipalities can apply to 1/3/5 kindergarten/s closest to their residence, which translates into *NA1/NA3/NA5*, i.e., availability based on the 1/3/5 nearest kindergartens.

Neighboring kindergarten availability is lower overall than local availability in both years. Using all three measures of neighboring availability, the average neighboring availability ranges from 0.05 to 0.30 (Table 1) and is lower when we assume that parents apply to more than one kindergarten (*NA5* is lower than *NA3* and that is lower than *NA1* in both years). Local kindergarten availability also decreased between 2001 and 2011, confirming the worsening availability of childcare in the Czech Republic (see the lower part of Table 1). The statistical distribution of neighboring kindergarten availability is presented in Figs 7–9. When we focus on the single nearest kindergarten (Fig 7), the resulting figure looks very similar to Fig 5 showing local availability: once again, large numbers of municipalities have capacity values of 0 and 1—meaning that, in the nearest neighboring municipality, there are either no available kindergarten places over and above those needed for children resident in that municipality or, on the contrary, that the nearest neighboring municipality has a kindergarten large enough to provide spaces for all children from neighboring villages who are not otherwise provided for locally. Figs 8 and 9 show far more frequent zero or low-capacity results for neighboring availability when the 3 or 5 nearest kindergartens are considered. This is because, if we suppose that

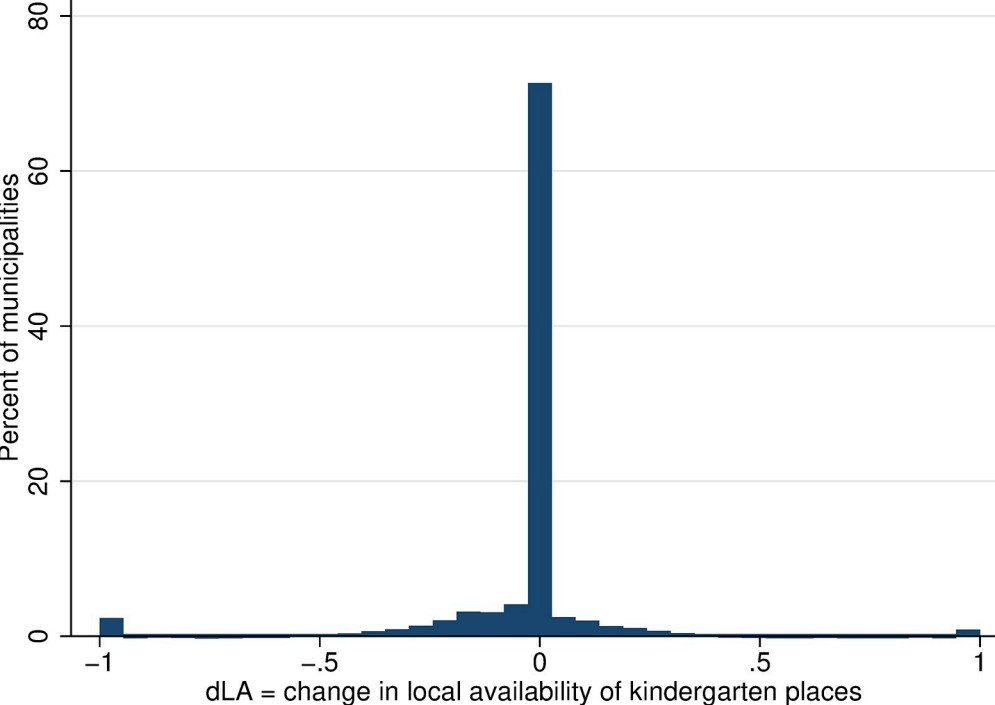

**Fig 6. Distribution of changes in local availability of kindergarten places.** Source: Census of Population, Buildings, and Dwellings; School Register (2001 and 2011), Author's calculations.

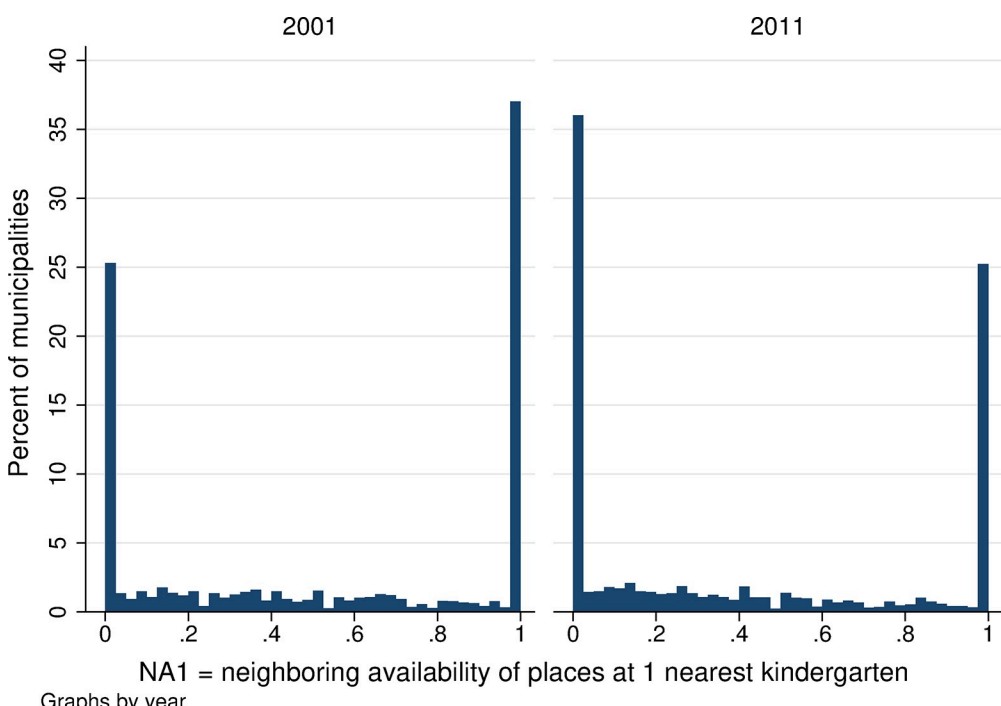

**Fig 7. Distribution of the availability of kindergarten places in neighboring areas, NA1.** Source: Census of Population, Buildings, and Dwellings; School Register (2001 and 2011), Author's calculations.

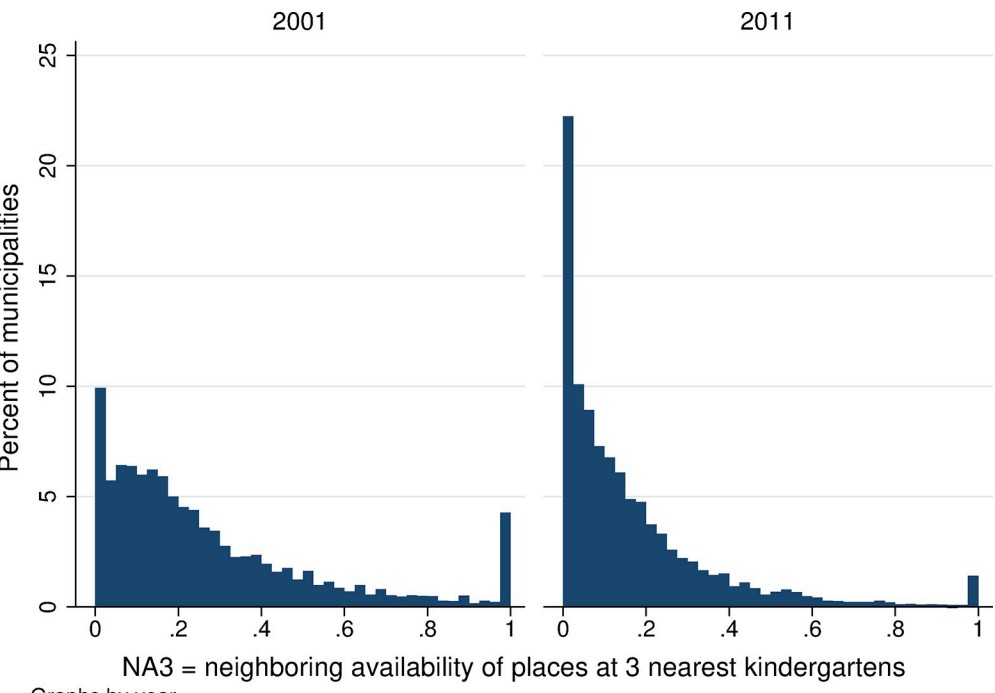

**Fig 8. Distribution of the availability of kindergarten places in neighboring areas, NA3.** Source: Census of Population, Buildings, and Dwellings; School Register (2001 and 2011), Author's calculations.

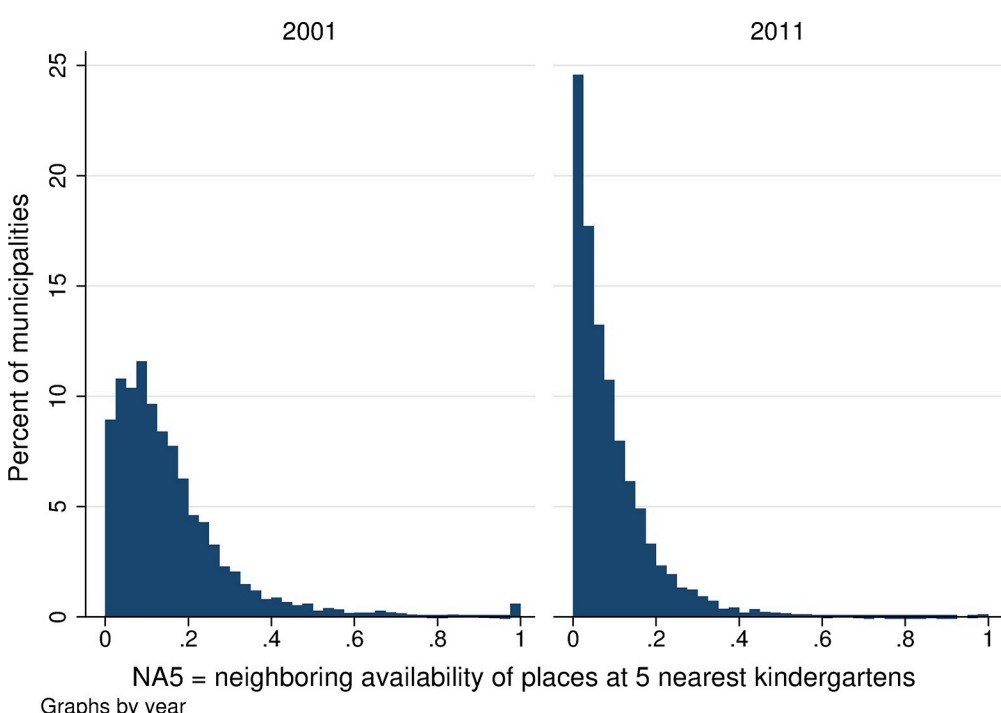

**Fig 9. Distribution of the availability of kindergarten places in neighboring areas, NA5.** Source: Census of Population, Buildings, and Dwellings; School Register (2001 and 2011), Author's calculations.

demand for the kindergarten in question is the sum of all not-otherwise-provided-for kindergarten-aged children from all municipalities for whom the given kindergarten is one of the nearest 3/5 kindergartens, it is highly unlikely that the given kindergarten would be able to meet the demand in full. Intuitively, this corresponds to the situation in which parents look for a kindergarten place within a broader radius of their home; while the chance of their child being admitted to each of these nearby kindergartens is small, the chance of being admitted to at least one of them is far higher.

The within-municipality changes in neighboring kindergarten availability were more substantial and more frequent than changes in local availability, and they were larger for the wider range of kindergartens considered (larger changes in Figs 10 and 11 than in Fig 12, which considers only the single nearest kindergarten). The distribution of all changes is predominantly in negative numbers, which indicates that the majority of mothers experienced a decrease in neighboring kindergarten availability between 2001 and 2011.

### 3.2. Regression model specification

We aim to estimate the causal impact of the (un)availability of kindergarten places on maternal employment using municipal-level data. However, there are various municipal-level characteristics that are likely to affect both the availability of childcare and maternal employment, such as the municipality's level of economic development and the age structure of the population. Therefore, using a simple OLS regression is not likely to provide causal estimates. To deal with this, we use a first differences model, which removes the effect of municipal-level characteristics that do not change over time and are possibly correlated with both childcare availability and maternal employment. This is a standard choice in the literature, which could be extended to a fixed effect model if we had more than two data points (see e.g., [16]). Unfortunately, data from the 2021 Census were not available at the time this manuscript was written.

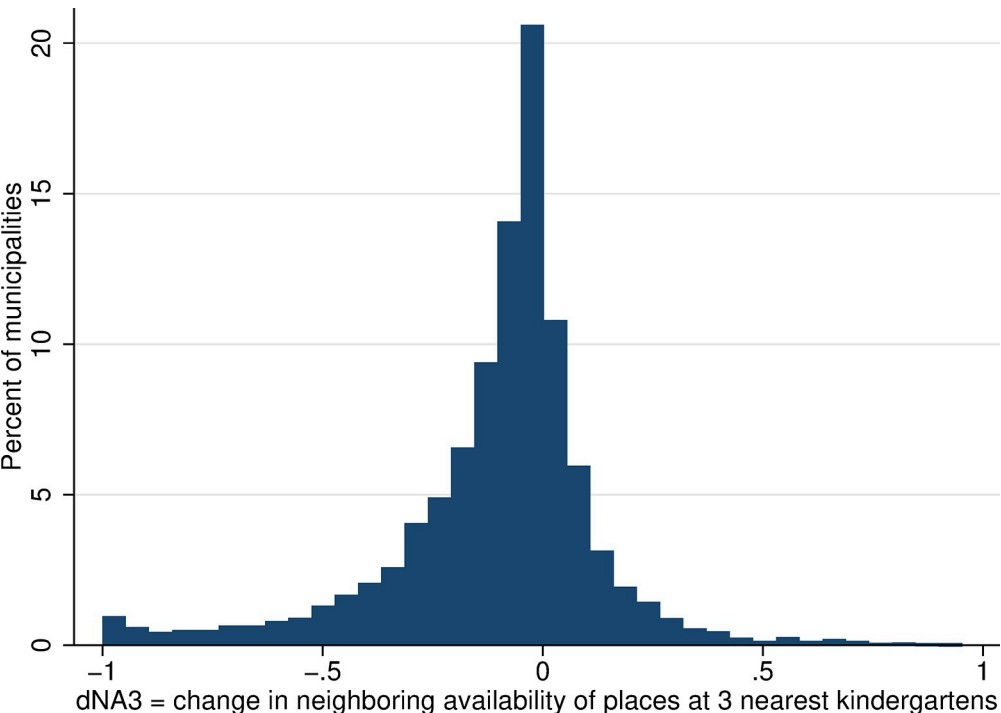

**Fig 10. Distribution of change in the availability of kindergarten places in neighboring areas, NA3.** Source: Census of Population, Buildings, and Dwellings; School Register (2001 and 2011), Author's calculations.

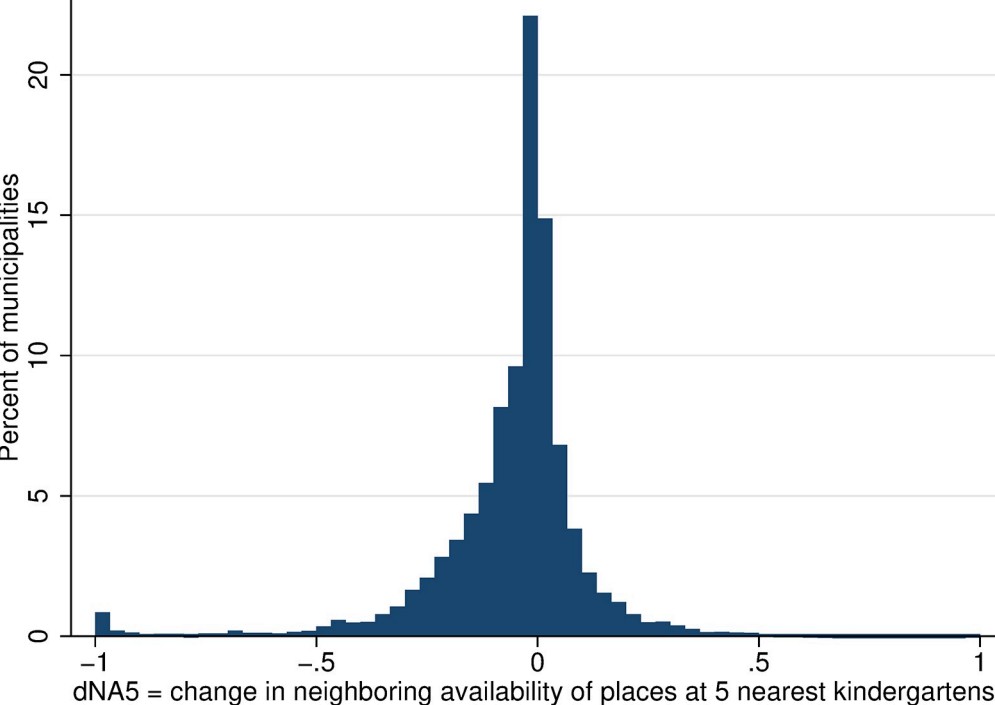

**Fig 11. Distribution of change in the availability of kindergarten places in neighboring areas, NA5.** Source: Census of Population, Buildings, and Dwellings; School Register (2001 and 2011), Author's calculations.

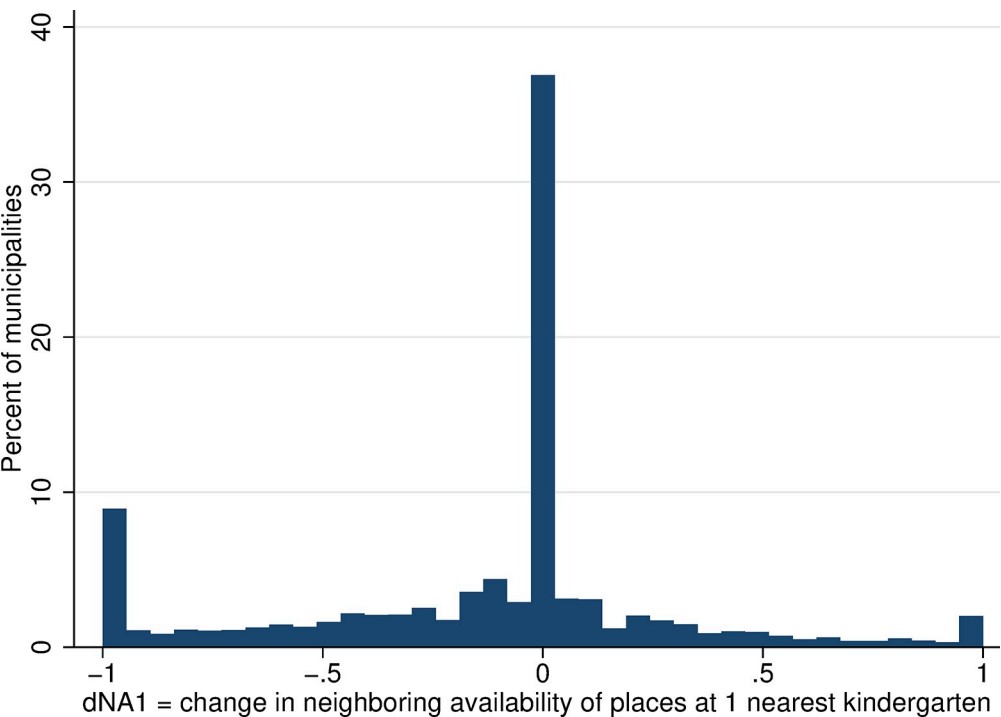

**Fig 12. Distribution of change in the availability of kindergarten places in neighboring areas, NA1.** Source: Census of Population, Buildings, and Dwellings; School Register (2001 and 2011), Author's calculations.

Therefore, our identification strategy is based on municipality-level changes in kindergarten availability over time that we assume are exogenous with respect to changes in the maternal employment rate. Therefore, we allow the municipal-level differences in kindergarten availability to be correlated with existing maternal employment rates, but we assume that the changes in kindergarten availability over time are orthogonal to changes in maternal employment. We discuss the plausibility of this assumption in Section 3.3.

Our baseline specification (**Specification 1.**) is the following:

$$\Delta EMPL_{i,d} = \boldsymbol{\alpha} + \boldsymbol{\beta_1}*\Delta LA_i + \boldsymbol{\beta_2}*\Delta NA_i + \boldsymbol{\gamma_1}*sec1_{i,d} + \boldsymbol{\gamma_2}*sec2_{i,d} + \boldsymbol{\gamma_3}*ter_{i,d} + \boldsymbol{\gamma_4}*children2_{i,d}$$
$$+ \boldsymbol{\gamma_5}*children3_{i,d} + X_{i,d} + \boldsymbol{\varepsilon_{i,d}}$$

where $\Delta EMPL_{i,d}$ represents the change in employment rates of mothers with a youngest child aged 3–5 between 2011 and 2001; $\Delta LA_i$ represents changes in local availability of kindergarten places between 2011 and 2011, $\Delta NA_i$ is changes in the neighboring availability of kindergarten places between 2011 and 2011 (in the estimation tables, $\Delta NA1$ shows availability based on the one nearest kindergarten, $\Delta NA3$ based on three, and $\Delta NA5$ based on five closest kindergartens). Among the control variables, we include indicator variables identifying the educational attainment of the given group of mothers (upper-secondary education without school leaving certificate–$sec1_{i,d}$; with school leaving certificate–$sec2_{i,d}$; tertiary education–$ter_{i,d}$). The reference group consists of mothers with incomplete upper-secondary education or lower. We also control for the number of children: $children2_{i,d}$ represents mothers with 2 children, $children3_{i,d}$ represents mothers with 3 and more children. In the regressions, the reference group is mothers with one child, i.e., $children1 = 1$.

Among other control variables ($X_{i,d}$), we include fixed effects for each district and for municipality population size categories (up to 500; 501–1,000; 1,001–2,000; 2,001–10,000; 10,001–100,000; 100,001–1 million, >1 million inhabitants). Municipality size is defined on the basis of 2011 data, so the values of these indicator variables do not change over time. These fixed effects provide for (that is, rule out the influence of) other district-specific characteristics and characteristics related to settlement size, which we cannot measure and that might also have influenced changes in the maternal employment rate.

The key parameters of our interest are therefore $\beta_1$ and $\beta_2$. They express the average change in the maternal employment rate in percentage points/100 caused by a change (in percentage points/100) in local or nearby availability of kindergarten places. The regression coefficients $\gamma_1$ to $\gamma_5$ for the other control variables capture autonomous changes in the employment rate over time for groups of mothers with different levels of education and numbers of children. These changes are unrelated to changes in kindergarten availability and are driven by other phenomena that we are not able to measure and include in this regression model.

We further extend the baseline model to allow for the impact of a kindergarten in the nearest neighboring municipality to be different if the municipality in question had no kindergarten (i.e., zero capacity) in both 2001 and 2011. Of the 6,098 municipalities in our sample, 2,789 did not have their own local kindergarten in either year (they had zero kindergarten capacity in both 2001 and 2011), so this is an important phenomenon. Table 2 provides a comparison

**Table 2. Summary statistics, municipalities with and without kindergartens.**

| | Employment rate among mothers | Average number of resident 3- to 5-year-olds | Share of 3- to 5-year-olds on total inhabitants | Share of municipalities with no 3- to 5-year-olds | Share of mothers with 3- to 5-year-old children by education | | | | Number of inhabitants | | | Number of municipalities |
|---|---|---|---|---|---|---|---|---|---|---|---|---|
| | | | | | primary | secondary without school leaving certificate | secondary with school leaving certificate | tertiary | average | min | max | |
| **Year 2001** | | | | | | | | | | | | |
| Municipalities with kindergartens in at least one year | 57.0% | 55.59 | 2.1% | 1.0% | 9.7% | 49.5% | 35.0% | 5.7% | 2 869 | 69 | 1 169 106 | 3 309 |
| Municipalities without kindergartens in any year | 52.3% | 5.06 | 2.0% | 5.1% | 11.3% | 51.7% | 32.4% | 4.6% | 259 | 26 | 85 855 | 2 789 |
| Total | 54.9% | 32.48 | 2.0% | 2.4% | 10.5% | 50.5% | 33.8% | 5.2% | 1 675 | 26 | 1 169 106 | 6 098 |
| **Year 2011** | | | | | | | | | | | | |
| Municipalities with kindergartens in at least one year | 67.0% | 50.93 | 1.8% | 0.2% | 7.0% | 34.6% | 46.4% | 12.0% | 2 913 | 52 | 1 268 796 | 3 309 |
| Municipalities without kindergartens in any year | 64.2% | 4.91 | 1.7% | 8.8% | 7.4% | 35.0% | 46.5% | 11.1% | 278 | 28 | 76 694 | 2 789 |
| Total | 65.8% | 29.88 | 1.7% | 4.1% | 7.2% | 34.8% | 46.4% | 11.6% | 1 708 | 28 | 1 268 796 | 6 098 |

Source: Census of Population, Buildings, and Dwellings; School Register (2001 and 2011), Author's calculations

of municipal characteristics between those that had a kindergarten in at least one of the two years ('municipalities with kindergartens'), and those that had no kindergarten in either year ('municipalities without kindergartens'). Municipalities with no kindergartens had, on average, ten times fewer inhabitants than municipalities with kindergartens, and registered inhabitants included, on average, 5 children between 3 and 5 years old, whereas in municipalities with kindergartens, the average number of officially resident 3- to 5-year-olds was substantially greater, at 56. The absence of kindergartens in small villages with low numbers of children is natural, but we allow for the possibility that neighboring kindergarten availability could be a more important factor in these municipalities. To do so, we introduce Specification 2.

**Specification 2.** is the following:

$$\Delta EMPL_{i,d} = \alpha + \beta_1 * \Delta LA_i + \beta_2 * \Delta NA_i + \beta_3 * \Delta NA_i x\, LA0_i + \gamma_1 * sec1_{i,d} + \gamma_2 * sec2_{i,d}$$
$$+ \gamma_3 * ter_{i,d} + \gamma_4 * children2_{i,d} + \gamma_5 * children3_{i,d} + X_{i,d} + \varepsilon_{i,d}$$

The extra term ($\Delta NA_i\, x\, LA0_i$), the interaction between changes in neighboring kindergarten availability levels ($\Delta NA_i$) and the indicator variable ($LA0_i$), which identifies municipalities with zero local kindergarten availability in both years (2001 and 2011), i.e., municipalities with no kindergarten in either period (zero capacity).

Finally, we extend our baseline specification to account for the fact that the impact of insufficient kindergarten capacity on mothers' employment may vary according to the mothers' levels of education. A natural factor at play here may be that better-educated mothers may have more options for finding paid childcare outside the public kindergarten network, such as at private or company kindergartens, with nannies, etc.

**Specification 3.** is the following:

$$\Delta EMPL_{i,d} = \alpha + \beta_{11} * pri_{i,d} * \Delta LA_i + \beta_{12} * sec1_{i,d} * \Delta LA_i + \beta_{13} * sec2_{i,d} * \Delta LA_i + \beta_{14} * ter_{i,d} * \Delta LA_i$$
$$+ \beta_2 * \Delta NA_i + \gamma_1 * sec1_{i,d} + \gamma_2 * sec2_{i,d} + \gamma_3 * ter_{i,d} + \gamma_4 * children2_{i,d}$$
$$+ \gamma_5 * children3_{i,d} + X_{i,d} + \varepsilon_{i,d}$$

This specification replaces terms involving the $\beta_1$ coefficient with terms involving the coefficients $\beta_{11}$ through to $\beta_{14}$, which reflect the impact of kindergarten availability on women's employment according to their education level (the coefficient $\beta_{11}$ represents mothers with primary education, $\beta_{12}$ to those with secondary education but without school leaving certificates, $\beta_{13}$ those with school leaving certificates, and $\beta_{14}$ mothers with tertiary education).

## 3.3. Identification assumptions

Our estimated regression coefficients can be interpreted as unbiased causal effects of changes in kindergarten availability on maternal employment only if the explanatory variables are not correlated with the stochastic term $\varepsilon_{i,d}$. Such correlations would exist if the changes in the overall maternal employment rate were driven by other changes not captured by our explanatory variables, which are themselves correlated with changes in local kindergarten availability. If there is a non-zero correlation, the resulting bias in the estimated coefficients depends on the magnitude of that correlation and the sign of the correlation determines the direction of the bias. As an example, consider a situation in which the availability of flex-time jobs increased more substantially over time in municipalities where kindergarten availability decreased the least. A greater availability of flexible working conditions would have a direct impact (not mediated via kindergarten availability) on the employment rate of mothers in those municipalities, and the fact that flex-time jobs were more concentrated in municipalities

with smaller changes in kindergarten availability would mean that our estimates of the impact of kindergarten availability on mothers' employment would be biased.

As discussed in Section 2, changes in regional availability of kindergarten places were driven by two main factors—demographic changes and changes in the number of kindergarten places (driven mostly by decisions made by municipal councils and their success in obtaining resources for kindergarten expansion). We assume that these changes are exogenous to the decisions of mothers with pre-school children concerning their return to the labor market. This would be violated if, for example, mothers who plan to return to the labor market sooner after childbirth pressured their municipal councils to increase kindergarten capacity. However, there are so many additional factors that can influence changes in kindergarten capacity even if the decision of a municipal council were affected by the preferences of local mothers, including a municipalities' success or failure attaining very competitive government kindergarten subsidies, demographic changes driven by construction of new houses or apartments, and migration decisions of families with young children, that we assume changes in kindergarten availability are mostly exogenous to mothers' labor market behavior.

Our key explanatory right-hand-side variables capturing kindergarten availability are also necessarily subject to measurement errors. Measurement error is likely to be caused by the fact that not all mothers are interested in sending their children to kindergarten. Additionally, our kindergarten availability indexes do not capture possible differences in priority entitlement to kindergarten places in neighbouring municipalities. Our measure of travel distance between neighbouring municipalities is only an approximation of the true time and financial demands of commuting, and we cannot account for the fact that some parents may prefer kindergartens that are close to their workplace, but further from their home. Anecdotal evidence does suggest that this is not very common, because children are eligible for child care places (if available) only in the municipality where they are registered as permanent residents. Further, delivering children to a pre-school facility and picking them up is frequently done by a different parent or by relatives. Those measurement errors very likely distort our key coefficient estimates towards zero. This means that the real effects of kindergarten unavailability on employment are likely larger than our estimates indicate. The extent of this bias is given by the variance in the measurement error relative to the variance in the real values of the variables. Naturally, this would also mean that the real losses to public finances, presented in Section 4.2, are also larger.

## 4. Results

### 4.1. Regression results

Table 3 presents *the* estimated coefficients of model specifications no. 1 in column (i) and no. 2 in column (iv), with the expanded versions of these specifications that test the sensitivity of the results to partial model modifications (columns ii, iii, v and vi). Estimates of specification no. 3, which allows for the possibility that insufficient kindergarten capacity may have a differential impact on employment depending on the mother's level of education, are presented in columns (vii)–(ix).

Source: Census of Population, Buildings, and Dwellings; School Register (2001 and 2011), Author's calculations.

### Impacts of local kindergarten availability

Our estimates for our key coefficient $\beta_1$ fall in the range 0.021–0.044 depending on the model specification used, and these estimates are statistically significant in all cases to at least the 5% level. This means that an increase in kindergarten availability by 10 percentage points led to a 0.21–0.44 p.p. increase in the employment rate of mothers of 3- to 5-year-olds. Given that the

**Table 3. Regression results.**

| | Specification 1 | | | Specification 2 | | | Specification 3 | | |
|---|---|---|---|---|---|---|---|---|---|
| | (i) | (ii) | (iii) | (iv) | (v) | (vi) | (vii) | (viii) | (ix) |
| $\Delta$LA | 0.0243*** | 0.0210*** | 0.0222*** | 0.0444*** | 0.0366*** | 0.0379*** | | | |
| | (0.00684) | (0.00666) | (0.00662) | (0.00776) | (0.00753) | (0.00748) | | | |
| $\Delta$LA x pri | | | | | | | 0.107*** | 0.104*** | 0.105*** |
| | | | | | | | (0.0289) | (0.0289) | (0.0289) |
| $\Delta$LA x sec1 | | | | | | | 0.0116 | 0.00824 | 0.00939 |
| | | | | | | | (0.0106) | (0.0105) | (0.0105) |
| $\Delta$LA x sec2 | | | | | | | 0.0306*** | 0.0274*** | 0.0285*** |
| | | | | | | | (0.00803) | (0.00788) | (0.00786) |
| $\Delta$LA x ter | | | | | | | -0.00397 | -0.00721 | -0.00611 |
| | | | | | | | (0.0134) | (0.0133) | (0.0132) |
| $\Delta$NA1 | -0.00321 | | | -0.00384 | | | -0.00321 | | |
| | (0.00251) | | | (0.00247) | | | (0.00251) | | |
| $\Delta$NA3 | | 0.00553 | | | 0.00883* | | | 0.00548 | |
| | | (0.00491) | | | (0.00484) | | | (0.00491) | |
| $\Delta$NA5 | | | 0.000671 | | | 0.00669 | | | 0.000611 |
| | | | (0.00852) | | | (0.00844) | | | (0.00852) |
| $\Delta$NA1 x LA0 | | | | -0.0134 | | | | | |
| | | | | (0.0200) | | | | | |
| $\Delta$NA3 x LA0 | | | | | -0.0870** | | | | |
| | | | | | (0.0367) | | | | |
| $\Delta$NA5 x LA0 | | | | | | -0.203*** | | | |
| | | | | | | (0.0689) | | | |
| Observations | 160 329 | 160 329 | 160 329 | 158 335 | 158 335 | 158 335 | 160 329 | 160 329 | 160 329 |
| $R^2$ | 0.023 | 0.023 | 0.023 | 0.024 | 0.024 | 0.024 | 0.023 | 0.023 | 0.023 |

Note: Dependent variable is the employment rate of mothers of 3–5 year-olds in all regressions. LA stands for local availability of kindergarten places, NA stands for availability of kindergarten places in neighboring areas. Robust standard errors in parentheses. Levels of statistical significance: *** $p<0.01$, ** $p<0.05$, * $p<0.1$

average employment rate of these mothers was 57.8% and the average availability of kindergarten places fell by 6.2 p.p., the estimated average effect corresponds to a relatively small decrease in the employment rate, from 57.8% to 57.7–57.5%. Nevertheless, these changes are relative and the overall impact on the number of mothers employed is determined by how large the changes in relative kindergarten availability were in smaller and larger municipalities. More specifically, the total real impact depends on how large the demographic groups of affected mothers are in relation to the magnitude of the change in relative kindergarten capacity. We present these calculations in the next section.

Based on our estimates of specification no. 3, the impact of kindergarten availability on women's employment is greatest for mothers with secondary school leaving certificates, for whom the estimated coefficient $\beta_1$ is between 0.027 and 0.031. We do not identify any significant effect for mothers with tertiary education, while for those with secondary education without school leaving certificates, and for mothers with only primary education, there is a statistically significant effect, but it is smaller than for mothers with secondary school leaving certificates.

The positive impact of kindergarten availability on employment that our estimates reveal for mothers of preschool-aged children is in line with most methodologically sound research

in other countries [3, 4, 6, 9, 23]. However, the magnitude of the effect we find is lower than existing estimates in studies on Central and Eastern European (CEE) countries, in which the impact of kindergarten childcare availability on mothers' employment is generally substantially higher than it is in Western European countries and the USA [15]. In CEE countries, an increase in the availability of kindergarten childcare of 10 p.p. has been linked to an increase in the employment rate of 1.2 p.p. (Hungary: [13]), or even as much as 4.2 p.p. (Poland: [14]). The effect we estimate is of a lower order, at just 0.2–0.4 p.p. We discuss these differences further in our conclusion.

### Impacts of neighboring kindergarten availability

Our estimates indicate that the availability of kindergarten places in neighboring municipalities does not have any significant impact, except in municipalities that had zero local kindergarten capacity in both years. However, these effects are negative, which is at odds with our hypothesis as to how nearby kindergarten availability might affect mothers' employment. One possible explanation for this finding is that the measure of neighboring availability we chose does not accurately reflect the reality of how parents search for kindergarten places within their surrounding area. For example, travel time by car may not be their main criterion, and parents may choose suitable kindergartens according to other criteria that we did not observe (e.g., information on the likelihood of being admitted, travel distance by public transport, etc.). The negative coefficient for kindergarten availability in the neighborhoods of municipalities that did not have their own kindergartens may be due to the fact that the number of children in the 3–5 age bracket increased in municipalities without kindergartens between 2001 and 2011, automatically reducing kindergarten availability in the surrounding area (because the number of resident 3- to 5-year-olds is in the denominator of the availability indicator) and, at the same time, the employment rate of mothers increased overall over the decade, but this increase was driven by other causes not observed in this analysis (e.g., greater availability of places in private kindergartens or increasing opportunities to work from home).

### 4.2. Aggregate effects

Our parametric estimates in section 4.1 represent percentage changes, without reference to the population sizes of the affected municipalities. In contrast, the country-level aggregation we present in this section does take the size of the affected populations into account. We use the parametric estimates from the previous section to quantify the overall impact of changes in kindergarten availability on maternal employment and the corresponding effect on public finances.

### Overall changes in kindergarten availability

According to Table 1, average kindergarten availability decreased in the period of interest–while in 2001, an average municipality provided kindergarten places for 82.9% of local children aged 3–5, it was only 76.6% in 2011. However, these figures fail to take two important facts into account. First, Czech municipalities vary widely in terms of population size (i.e., the number of children affected) and this is not accounted for in these averaged figures for kindergarten availability. Second, kindergarten availability is a local phenomenon: a surplus of available kindergarten places in one municipality does not necessarily remedy a lack of availability in another municipality. It would thus be misleading to derive aggregate figures for kindergarten availability from a country-wide comparison of total kindergarten capacity against the total number of 3- to 5-year-olds.

Instead, we estimate the aggregate *unavailability* of kindergarten places as the sum of the differences between the number of 3- to 5-year-olds resident in a given municipality and kindergarten capacity in that municipality, counting *only municipalities in which there was insufficient capacity*. The deficit in local kindergarten capacity, defined in this way, increased substantially, by two thirds, from 63,027 places lacking in 2001 to 105,771 in 2011. Obviously, some parents may have placed their children in kindergartens in neighboring municipalities. Neighboring availability also decreased slightly between the years observed. Specifically, in 2001, 15,469 children for whom kindergarten places were not available in their own municipalities would have found places at kindergartens in neighboring municipalities, but this number fell to 14,252 in 2011. Capacity provided by nearby kindergartens reduces the estimated aggregate lack of kindergarten places to 47,558 places in 2001 and to 91,519 places in 2011. Both approaches (using only local or also neighboring kindergartens) suggest that the increase in the deficit of kindergarten places between 2001 and 2011 was huge–an additional 43 thousand children did not find a place in 2011 compared to the numbers in 2001. This represents approximately 36% of the cohort of 3-year-old children in 2011.

## Overall changes in employment

The data on changes in employment rates of mothers with preschool-aged children in Table 1 document the situation in the average Czech municipality. In reality, municipalities vary in terms of the number of mothers affected and the values of the other explanatory variables. We therefore calculate the aggregate effect of changes in local kindergarten availability on mothers' employment as the sum of the changes in absolute employment as a result of changes in local kindergarten availability, aggregated across all municipalities. We work on the basis of the definition of our key indicator of change in the employment rate

$$\Delta EMPL_{i,d} = \frac{E_{i,d}^{2011}}{P_{i,d}^{2011}} - \frac{E_{i,d}^{2001}}{P_{i,d}^{2001}},$$

where $E_{i,d}^{2001}$ and $E_{i,d}^{2011}$ and $P_{i,d}^{2001}$ and $P_{i,d}^{2011}$ represent the total numbers of employed mothers, specifically those in the demographic group $d$ in municipality $i$ in both years. Because the effect expressed by the estimated coefficient $\beta$ is given in percentage points as

$$\Delta EMPL_{i,d} = \hat{\beta} * \Delta LA_{i,d}$$

the impact on the total number of employed mothers in the given demographic group in the given municipality can be calculated as

$$E_{i,d}^{2011} - E_{i,d}^{2001} \equiv \Delta E_{i,d} = \left[\hat{\beta}\Delta LA_{i,d} + \frac{E_{i,d}^{2001}}{P_{i,d}^{2001}}\right] P_{i,d}^{2011} - E_{i,d}^{2001},$$

where the term in brackets represents our predicted employment rate in 2011. By adding together the values of $\Delta E_{i,d}$ for all municipalities, we obtain the total change in the number of employed mothers caused by changes in kindergarten availability. This impact on total employment is therefore clean of other factors which, in parallel, surely also affected changes in the employment rate between the two years.

Our estimates of the net aggregate impact of kindergarten availability on mothers' employment, corresponding to our various model specifications, are presented in row (a) of Table 4. Overall, the lack of available kindergarten places resulted in between 8,847 and 9,111 fewer mothers being employed, according to the model specification we use. That is between 4.9% and 5.0% of all mothers whose youngest child was between 3 and 5 years old in 2011.

**Table 4. Aggregate changes in employment and fiscal impacts.**

| | | Specification 1 | | | Specification 2 | | |
|---|---|---|---|---|---|---|---|
| | | (i) | (ii) | (iii) | (iv) | (v) | (vi) |
| | β (coefficient for the effect of LA on the employment rate) | 0.0243*** | 0.0210*** | 0.0222*** | 0.0444*** | 0.0369*** | 0.0383*** |
| (a) | Predicted change in maternal employment: | | | | | | |
| | • in absolute number | 8 884 | 8 847 | 8 861 | 9 111 | 9 026 | 9 042 |
| | • as a share of all mothers with children 3–5 | 4.87% | 4.85% | 4.86% | 5.00% | 4.95% | 4.96% |
| (b) | Predicted change in maternal employment (alternative calculation not considering the population size of municipalities): | | | | | | |
| | • in absolute number | 9 352 | 9 313 | 9 327 | 9 585 | 9 498 | 9 514 |
| | • as a share of all mothers with children 3–5 | 5.13% | 5.11% | 5.12% | 5.26% | 5.21% | 5.22% |
| (c) | Net fiscal impact of reduced maternal employment (in billion CZK per year): | | | | | | |
| | • benchmark scenario | 1.652 | 1.645 | 1.647 | 1.694 | 1.678 | 1.681 |
| | • representative-sample scenario | 1.185 | 1.180 | 1.182 | 1.215 | 1.204 | 1.206 |

Note: Statistical significance of coefficient point estimates: *** $p<0.01$, ** $p<0.05$, * $p<0.1$

If we did not take into account the correlation between the population size of a municipality (number of resident mothers) and the degree of kindergarten unavailability, we could calculate the aggregate impact more simply, based on the above equations, using the average values for the country as a whole. These estimates are given in line (b) of Table 4. These values are somewhat higher than those in line (a), which indicates that the reduction in kindergarten availability was more highly concentrated in municipalities with smaller populations.

## Impact on public finances

Line (c) of Table 4 enumerates the gross annual impact on public finances associated with the lower maternal employment rate in 2011 due to the greater unavailability of kindergarten places. For this calculation, we made use of fiscal impact parameter estimates from [24]. [24] estimated the net fiscal loss to the public purse as the difference between the total revenues from one additional kindergarten place and the costs of providing such a place. On the revenue side, the study primarily considers additional incomes to public budgets via more income tax payments and social security contributions, resulting from the mother's employment, and reduced social benefit payments to the family. The main costs would be the additional investment required to create and run an additional kindergarten place. In that study, we calculate the fiscal loss connected to the absence of mothers of pre-school children on the labor market. This loss includes foregone income taxes, and social security and health insurance contributions, and additional expenditures on social benefits.

We consider two scenarios: (i) *a benchmark scenario* where all mothers with pre-school kids returns to the labor market at an average female wage and (ii) *a representative-sample scenario*, which imputes wages to a representative sample of non-working mothers of pre-school children based on their characteristics and corrects for selection into employment (for details, see [24]). The first scenario estimates the fiscal loss of forgone taxes at about 158,000 CZK per year and additional expenditures on social benefits of 28,000 CZK per year for each woman

who is out of work due to insufficient childcare capacities. The second scenario estimates the forgone taxes to be about 123,000 CZK and expenditures on social benefits of about 10,000 CZK per year (1 Euro equals about 25 Czech crowns, CZK). [24] also provide several robustness checks and sensitivity analyses of these estimates. If anything, the numbers reported above are a lower bound of the true financial loss caused by the absence of mothers of preschool children on the labor market. Among other things, accounting for the fact that earlier returns to work would increase the total lifelong earnings of women and thus further increases the estimated fiscal loss by additional 53,000 CZK per each woman. The calculation of these secondary effects is based on the finding that every additional year of absence from the labor market decreases life-long earnings of Czech women by about 1% [25].

The reduction in kindergarten availability between 2001 and 2011 that we observe here, which caused a decrease in employment affecting about 9,000 mothers, thus corresponds to a net loss to the public purse of about 1.2 billion CZK per year, using the representative-sample scenario, and about 1.7 billion CZK per year based on the benchmark scenario (Table 4). If we also add losses caused by knock-on negative effects on affected mothers' future earnings via secondary effects, this loss would be a little higher, between 1.4 billion and 1.9 billion CZK per year. This represents about 0.1–0.15% of the state budget revenues.

The total loss to the public budgets was, of course, accumulated over the entire period of 2001–2011. If the reduction in kindergarten capacity had occurred all at once in 2002, the total net loss would have been 12–17 billion CZK or, if we take secondary effects into account, 14–19 billion CZK.

We must not overlook two more sources of additional losses: (a) the financial losses quantified above only relate to the change (decrease) in kindergarten capacity between 2001 and 2011. At the beginning of the period, there was an existing shortage of kindergarten places, which was not resolved and which caused additional budget losses during the period we analyze; (b) the overall change in the employment rate among mothers in 2001–2011 was affected not only by the reduction in kindergarten capacity, but also by other effects unknown to us and not included in our model. We are unable to calculate the net losses to public finances resulting from them, because the potential costs of measures to counteract these factors are not known, as the costs of additional kindergarten places are.

## 5. Discussion and conclusion

This paper evaluates the effects of local (un)availability of kindergarten places on the employment rates of mothers with preschool-aged children. Our study uses municipal-level and time variation in kindergarten availability to identify the causal effect on maternal employment. We combine partially aggregated data from the Census of Population, Buildings, and Dwellings, conducted by the Czech Statistical Office in 2001 and 2011, with data on the capacity of public kindergartens at the municipality level taken from the School Register maintained by the Ministry of Education, Youth, and Sports of the Czech Republic into a unique database. We also take advantage of travel distance data to calculate kindergarten availability not only in a specific municipality, but also in its closest neighboring kindergartens.

We contribute to scarce literature on the impact of childcare availability on maternal employment in Central and Eastern European (CEE) countries. The few existing studies from CEE countries [13–15] found a substantially higher impact of childcare on maternal employment than what was found in previous studies in Western Europe and the US. Previous studies explained this phenomenon by referring to a specific institutional setting in CEE countries that leads to low maternal employment, but overall high female labor force participation, and thus provides substantial room for raising maternal employment rates. However, the existing

evidence for Central and Eastern Europe is so far very limited, and we contribute to this literature by providing estimates from the Czech Republic.

We estimate the impact of the availability of public kindergarten places on employment of mothers of 3- to 5-year-olds using unique municipal-level variation in the availability of kindergarten places stemming from two sources. First, we utilize notable dynamic demographic changes in terms of non-uniform distribution of children born across municipalities. Second, we take advantage of the fact that the Czech Republic has an extraordinarily large number of self-governed municipalities, which are the primary kindergarten providers. We argue that changes in the availability of kindergarten places at the municipality level can thus be considered exogenous in our setting.

Our estimates imply that an increase in public kindergarten availability of 10 percentage points leads to an increase in the employment rate of mothers of 3- to 5-year-olds of 0.2–0.4 percentage points. These results are much lower in magnitude than what was found in previous studies from Hungary and Poland, where increasing childcare coverage by 10 p.p. increases maternal employment by 1.17 p. p. [13] and 4.2 p.p. [14]. Our results are much more comparable to studies from non-CEE countries that also found relatively modest effects.

Why are our results quantitatively different from estimates in other CEE countries? It could be that the limited evidence to date from Central European countries does not capture the underlying differences between these countries. While they are similar in some institutional settings (e.g., the low capacity of public childcare for children under 3 years and limited flexible working arrangements), they naturally differ in many other dimensions that are important for maternal employment, such as the length of maternity and parental leaves and gender norms (see [26], for a discussion). Quantitative differences in the size of estimated effects could also stem from differences in estimation strategies used in different studies (see e.g., [27], for a discussion of how different estimation methods can affect estimated labor supply elasticities). To the best of our knowledge, there is only one study that provides estimates of the causal impact of childcare availability on maternal employment in the Czech Republic [15], and their estimation strategy is based on childcare eligibility cutoffs, comparing children who reach the age of 3 around the eligibility cut-off. Their estimates for the Czech Republic are similar in magnitude to those from Hungary (using the same estimation strategy), suggesting that the estimation strategy is a more important determinant than institutional differences between the two countries. However, their study differs from ours not only in its estimation strategy, but also in their definition of the treatment group. While their results apply only to mothers with a 3-year-old child born close to the childcare eligibility cut-off point, our estimates apply a much broader group of mothers with a youngest child aged 3–5.

Even though our measure of childcare availability defined as a ratio of available places to eligible children is commonly used in the literature [11, 14, 16, 22], it probably suffers from measurement errors. Measurement error in local kindergarten availability might be caused by the fact that not all mothers are interested in sending their 3- to 5-year-old children to kindergarten. For neighboring kindergarten availability, measurement error might also stem from the unobserved differences in priority entitlement to kindergarten places in neighboring municipalities and the imprecision of our measure of travel distance, which is only an approximation of the true time and financial costs of commuting. While this is a limitation of our research, measurement error alone cannot explain the differences in existing evidence from CEE countries. [14] use the very same measure as we do for Poland and arrive at a much higher estimated elasticity of maternal labor supply, while [15] use an individual-level eligibility cut-off which does not suffer from this kind of measurement error, and they estimate a much higher elasticity than we do (but lower than [14]).

Another limitation of our research lies in the identification assumption of the first differences method we use. We assume that changes in kindergarten availability are exogenous to the decision of mothers with pre-school children concerning their return to the labor market. This could be violated if changes in childcare availability are driven by the demand side, which would then raise the potential for reverse causality and estimation bias (for a detailed discussion, see e.g., [22]). Similarly to [22], we argue that was likely not the case in the Czech Republic. The substantial number of rejected applications for kindergartens that continued to grow over time suggests a substantial excess demand, i.e., variation in childcare availability in our setting likely stems from supply side and demographic changes.

## Author Contributions

**Conceptualization:** Klára Kalíšková, Daniel Münich.

**Formal analysis:** Klára Kalíšková.

**Methodology:** Klára Kalíšková.

**Software:** Klára Kalíšková.

**Supervision:** Daniel Münich.

**Visualization:** Klára Kalíšková.

**Writing – original draft:** Klára Kalíšková.

**Writing – review & editing:** Klára Kalíšková, Daniel Münich.

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
