## [Decision Letter · Decision Letter 0]

15 Mar 2023

PONE-D-22-33710The Impact of Childcare Availability on Maternal Employment: Evidence from Czech MunicipalitiesPLOS ONE

Dear Dr. Kalíšková,

Thank you for submitting your manuscript to PLOS ONE. After careful consideration, we feel that it has merit but does not fully meet PLOS ONE’s publication criteria as it currently stands. Therefore, we invite you to submit a revised version of the manuscript that addresses the points raised during the review process.

The reviewers have provided very useful comments and suggestions for you to revise the article. Please revise following their comments. One of the reviewers also comments about the data being out of date. Could you please explain in the article why more up to date have not been used: are these not available? If they are available but you have not used them, you will have to justify your choice using some persuasive arguments.

We look forward to receiving your revised manuscript.

Kind regards,

Daphne Nicolitsas

Academic Editor

PLOS ONE

3. We note that Figures 4 and 5 in your submission contain [map/satellite] images which may be copyrighted. All PLOS content is published under the Creative Commons Attribution License (CC BY 4.0), which means that the manuscript, images, and Supporting Information files will be freely available online, and any third party is permitted to access, download, copy, distribute, and use these materials in any way, even commercially, with proper attribution. For these reasons, we cannot publish previously copyrighted maps or satellite images created using proprietary data, such as Google software (Google Maps, Street View, and Earth). For more information, see our copyright guidelines: http://journals.plos.org/plosone/s/licenses-and-copyright.

a. You may seek permission from the original copyright holder of Figures 4 and 5 to publish the content specifically under the CC BY 4.0 license. 

Reviewers' comments:

Reviewer's Responses to Questions

**Comments to the Author**

1. Is the manuscript technically sound, and do the data support the conclusions?

Reviewer #1: Yes

Reviewer #2: Yes

2. Has the statistical analysis been performed appropriately and rigorously? 

Reviewer #1: Yes

Reviewer #2: Yes

3. Have the authors made all data underlying the findings in their manuscript fully available?

Reviewer #1: Yes

Reviewer #2: No

4. Is the manuscript presented in an intelligible fashion and written in standard English?

Reviewer #1: Yes

Reviewer #2: Yes

5. Review Comments to the Author

Reviewer #1: Dear authors,

I am glad I have this opportunity to review your manuscript. My comments are as follows (let me divide my comments according to the structure of your manuscript):

Abstract

- I personally expect that the aim of the study will be exactly defined, but it is missing in the current version

- identify "various sources" you are using for your research

- data used are more than 10 years old, do you consider it as still actual?

- add more information about the methodology (you are presenting partial results without information about the way of their processing/achieving)

- be more specific (...is lower than...; is closer to...)

Introduction

- the topic is introduced at a good level by using actual literature

Institutional background

- additional information about the situation in the Czech Republic is presented at an expected level

Methodology

- selection of indicators should be explained (you are working with local kindergarten availability and

neighboring kindergarten availability, but why? what kind of indicators are used to capture the availability of kindergartens? show other research/indicators used for a similar purpose to declare what is new in your research

- add some numeric information about both indicators used (at least describe them by some moment characteristics)

- the same problem with the regression model, any explanation for choosing this method is missing (I agree with your choice, but I also expect much more information)

Results

- all tables/figures in section 4.1 are described very well, but there is no idea throughout the whole text

- each paragraph starts "XXX presents something...", after this paragraph you are not working with achieved information

- section 2.1 is much better prepared, the description of the results is divided and interpreted in detail

- an indicator of change in the employment rate and other parameters/indicators should be described in the methodology, not in the results (section 4.3)

Discussion and conclusion

- add information about the limitations of your research

Formal notes

- add the name of the axis Y (Figure 1, Figure 2)

I wish you all the best with this manuscript and other ones in the future

Reviewer #2: The paper provides estimates of the impact of local public kindergarten availability on the employment rate of Czech mothers of pre-school children. The authors use unique datasets from the Census carried out by the Czech Statistical Office in 2001 and 2011 matched with the data on the capacity of kindergartens in Czech municipalities from the Ministry of Education, Youth and Sports. The estimates are lower than those published previously for the Czech Republic, although the evidence is limited and the methodology is different, while they are comparable to previous estimates for other Southern and Western European countries. In the final part the authors calculate the aggregate effects and fiscal impacts.

The paper is definitely technically sound. I appreciate the part explaining identification assumptions and the discussion in the concluding section on why the estimates are different from results for other countries in the region. I see the discussion on why the estimates are different as the main challenge. I discuss below some related points.

First, the measurement errors described on p. 12 distort the coefficient estimates towards zero. Are the measurement errors responsible for at least part of the difference between their results and those in Lovász and Szabó-Morvai (2017)? I understand the main difference is in methodology as Lovász and Szabó-Morvai (2017) rely on childcare eligibility cutoffs comparing children that reach the age of 3 around the eligibility cuttoff.

Second, the generous parental leave in the Czech Republic (pp. 4-5) allows mothers to stay with their children at home beyond 3 years of age, while the state support is less generous in other countries. Could this explain why the estimates are lower?

Third, the Czech Republic has very large number of self-governed municipalities which are the main providers of kindergarten places. I suspect the typical commuting distance to potential work could be longer than within municipalities and mothers take advantage of kindergartens available near their workplaces. Is it relevant? Considering different level of aggregation could be left for future research.

Finally, are the aggregate effects large or small? The calculation relies on parameters and other estimates from Kalíšková et al. (2016). I would appreciate some discussion of robustness of the calculation. I suggest to elaborate on it and mention the aggregate effects in the abstract or remove this part.

Overall, I enjoyed reading the paper. My comments are meant to improve the discussion related to the comparison with previous literature as well as the robustness of their calculation of aggregate impacts.

6. PLOS authors have the option to publish the peer review history of their article (what does this mean?). If published, this will include your full peer review and any attached files.

Reviewer #1: **Yes: **Roman Vavrek

Reviewer #2: No

---

## [Author Response · Author response to Decision Letter 0]

2 Jun 2023

We would like to thank the two reviewers for very constructive and useful comments. We respond to their comments one by one in an enclosed letter to reviewers titled "Response to Reviewers" and we also summarize the major changes we made to the manuscript in the Cover letter. Thank you once again for considering our paper for publication in PLOS ONE.

---

## [Decision Letter · Decision Letter 1]

10 Jul 2023

The Impact of Childcare Availability on Maternal Employment: Evidence from Czech Municipalities

PONE-D-22-33710R1

Dear Dr. Kalíšková,

We’re pleased to inform you that your manuscript has been judged scientifically suitable for publication and will be formally accepted for publication once it meets all outstanding technical requirements.

Kind regards,

Daphne Nicolitsas

Academic Editor

PLOS ONE

Additional Editor Comments (optional):

Reviewers' comments:

Reviewer's Responses to Questions

**Comments to the Author**

1. If the authors have adequately addressed your comments raised in a previous round of review and you feel that this manuscript is now acceptable for publication, you may indicate that here to bypass the “Comments to the Author” section, enter your conflict of interest statement in the “Confidential to Editor” section, and submit your "Accept" recommendation.

Reviewer #1: All comments have been addressed

Reviewer #2: All comments have been addressed

2. Is the manuscript technically sound, and do the data support the conclusions?

Reviewer #1: Yes

Reviewer #2: Yes

3. Has the statistical analysis been performed appropriately and rigorously? 

Reviewer #1: Yes

Reviewer #2: Yes

4. Have the authors made all data underlying the findings in their manuscript fully available?

Reviewer #1: Yes

Reviewer #2: Yes

5. Is the manuscript presented in an intelligible fashion and written in standard English?

Reviewer #1: Yes

Reviewer #2: Yes

6. Review Comments to the Author

Reviewer #1: Dear authors,

thank you for taking the time to incorporate my previous comments/notes, much appreciated. Let me process this review in the structure of the first one:

Abstract

- I personally expect that the aim of the study will be exactly defined, but it is missing in the current version

validation: partially incorporated

- identify "various sources" you are using for your research

validation: done (the term is erased)

- data used are more than 10 years old, do you consider it as still actual?

validation: explained

- add more information about the methodology (you are presenting partial results without information about the way of their processing/achieving)

validation: incorporated

- be more specific (...is lower than...; is closer to...)

validation: incorporated

Introduction

- the topic is introduced at a good level by using actual literature

validation: without the required modifications

Institutional background

- additional information about the situation in the Czech Republic is presented at an expected level

validation: without the required modifications

Methodology

- selection of indicators should be explained (you are working with local kindergarten availability and

neighboring kindergarten availability, but why? what kind of indicators are used to capture the availability of kindergartens? show other research/indicators used for a similar purpose to declare what is new in your research

- add some numeric information about both indicators used (at least describe them by some moment characteristics)

- the same problem with the regression model, any explanation for choosing this method is missing (I agree with your choice, but I also expect much more information)

validation: incorporated

Results

- all tables/figures in section 4.1 are described very well, but there is no idea throughout the whole text

- each paragraph starts "XXX presents something...", after this paragraph you are not working with achieved information

- section 2.1 is much better prepared, the description of the results is divided and interpreted in detail

- an indicator of change in the employment rate and other parameters/indicators should be described in the methodology, not in the results (section 4.3)

validation: incorporated

Discussion and conclusion

- add information about the limitations of your research

validation: incorporated

Formal notes

- add the name of the axis Y (Figure 1, Figure 2)

validation: incorporated

I wish you many citations of this manuscript around the world.

Reviewer #2: The authors addressed all my comments and questions. In particular, I appreciate the improved comparison with the previous literature and the robustness of their calculation of aggregate effects.

7. PLOS authors have the option to publish the peer review history of their article (what does this mean?). If published, this will include your full peer review and any attached files.

Reviewer #1: **Yes: **Roman Vavrek

Reviewer #2: No

---

## [Editor Report · Acceptance letter]

13 Jul 2023

PONE-D-22-33710R1 

The impact of childcare availability on maternal employment: Evidence from Czech municipalities 

Dear Dr. Kalíšková:

I'm pleased to inform you that your manuscript has been deemed suitable for publication in PLOS ONE. Congratulations! Your manuscript is now with our production department. 

Kind regards, 

on behalf of

Dr. Daphne Nicolitsas 

Academic Editor

PLOS ONE